# 🐺 Wolf: Dense Video Captioning with a World Summarization Framework

**Boyi Li**[1,2]     **Ligeng Zhu**[1,3]     **Ran Tian**[1,2]     **Shuhan Tan**[1,4]
**Yuxiao Chen**[1]     **Yao Lu**[1]     **Yin Cui**[1]     **Sushant Veer**[1]     **Max Ehrlich**[1]
**Jonah Philion**[1,5]     **Xinshuo Weng**[1]     **Fuzhao Xue**[1]     **Linxi Fan**[1]     **Yuke Zhu**[1,4]
**Jan Kautz**[1]     **Andrew Tao**[1]     **Ming-Yu Liu**[1]     **Sanja Fidler**[1,5]     **Boris Ivanovic**[1]
**Trevor Darrell**[2]     **Jitendra Malik**[2]     **Song Han**[1,3]     **Marco Pavone**[1,6]
[1]**NVIDIA**  [2]**UC Berkeley**  [3]**MIT**  [4]**UT Austin**  [5]**University of Toronto**  [6]**Stanford University**

**Reviewed on OpenReview:** `https://openreview.net/forum?id=Z1dH7hao7p`

## Abstract

We propose **Wolf**, a WOrLd summarization Framework for accurate video captioning. Wolf is an automated captioning framework that adopts a mixture-of-experts approach, leveraging complementary strengths of Vision Language Models (VLMs). By combining image and video models, our framework captures different levels of information and summarizes them efficiently. Our approach can be applied to enhance video understanding, auto-labeling, and captioning. To evaluate caption quality, we introduce CapScore, an LLM-based metric to assess the similarity and quality of generated captions compared to the ground truth captions. We further build four human-annotated datasets in three domains: autonomous driving, general scenes, and robotics, to facilitate comprehensive comparisons. We show that Wolf achieves superior captioning performance compared to state-of-the-art approaches from the research community (VILA-1.5, CogAgent) and commercial solutions (Gemini-Pro-1.5, GPT-4V). For instance, in comparison with GPT-4V, Wolf improves CapScore both quality-wise by 55.6% and similarity-wise by 77.4% on challenging driving videos. Finally, we establish a benchmark for video captioning and introduce a leaderboard, aiming to accelerate advancements in video understanding, captioning, and data alignment.

## 1 Introduction

Video captioning is crucial as it facilitates content understanding and retrieval by providing accurate, searchable descriptions. It also provides pairwise data for effective training of foundation models for tasks like video generation, such as Sora (Brooks et al., 2024), Runaway (Runway, 2024) and Wan2.1 Team (2025) . However, generating descriptive, accurate, and detailed video captions remains a challenging research problem for several reasons: *firstly*, high-quality labeled data are scarce. Video captions from the internet can be faulty and misaligned and human annotation is prohibitively expensive for large datasets. *Secondly*, video captioning is inherently more challenging than image captioning due to the additional complexity of temporal correlation and camera motion. Existing captioning models (Hong et al., 2024; Zhang et al., 2023) struggle with temporal reasoning and fail to achieve accurate scene understanding. *Thirdly*, there is no established benchmark to measure captioning progress. Existing video QA benchmarks (Maaz et al., 2023) are often limited to short answers, making it difficult to measure hallucinations in detailed long captions. Fourthly, the correctness and completeness of the captions are crucial for safety-critical tasks. In the era of large language models (LLMs), text descriptions of scenarios used by embodied agents for planning and control become increasingly common (Mao et al., 2023a;b; Li et al., 2024; Ding et al., 2023). Consequently, a false or incomplete description of the scenario may lead to the decision-making module overlooking a critical object after training on such caption data, resulting in safety risks. For instance, missing the presence of a human in the vicinity of a vegetable-chopping manipulator can lead to an injury.

To handle these challenges, we introduce WOrLd summarization Framework (**Wolf**), a novel summarization captioning framework, along with a captioning metric CapScore, and the Wolf captioning benchmark with corresponding datasets. Unlike previous works that utilize a single model to generate captions, we propose to use multiple models to collaborate (Jiang et al., 2024), producing much more accurate captions. By leveraging multiple models, we can provide more fine-grained details while reducing hallucinations. We show that Wolf achieves superior captioning performance compared to state-of-the-art approaches from the research community (such as VILA (Lin et al., 2023c), CogAgent (Hong et al., 2024)) to commercial solutions (such as Gemini-Pro-1.5 (Team et al., 2023), GPT-4V (OpenAI, 2023)). In summary, we have three main contributions:

1. We design the first world summarization framework **Wolf** for video captioning and introduce an LLM-based metric **CapScore** for evaluating the quality of captions. We have further verified that CapScore aligns with human evaluations and is more effective than several widely used captioning metrics. The results show that our method improves CapScore by a large margin.

2. We introduce four benchmark datasets. These datasets include autonomous driving, general scenes from Pexels, and robotics videos, along with human-annotated captions, referred to as the **Wolf Dataset**.

3. The code, data and benchmark has been been release in https://wolfv0.github.io/. Continuous efforts and improvements will be made to refine the Wolf Dataset, codebase, and CapScore. We hope that Wolf will raise awareness about the quality of video captioning, set a standard for the field, and boost community development.

## 2 Related Works

**Image Captioning.** Visual language models (VLMs) have shown rapid advancements, achieving leading performance in image captioning tasks, largely due to the success of LLMs. CLIP (Radford et al., 2021) pioneered this field by training a shared feature space for vision and language modalities on image-caption pairs. Building on CLIP, BLIP (Li et al., 2022) and BLIP-2 (Li et al., 2023) improved performance by aligning the pre-trained encoder with LLMs. Following the direction, LLaVA (Liu et al., 2023) and InstructBLIP (Dai et al., 2023) demonstrated that jointly training on diverse datasets as an instruction-following task leads to strong generalization across various tasks. VILA (Lin et al., 2023c) highlighted the importance of pre-training with diverse data, and therefore significantly scaled up the pre-training dataset. Kosmos-2 (Peng et al., 2023) and PaLI-X (Chen et al., 2023b) further introduced pseudo-labeling bounding boxes from open-vocabulary object detectors to scale up the size of pre-training dataset.

**Video Captioning.** As image-based VLMs are not trained with video data, they are limited in describing details present in the video data (Zhou et al., 2024; Kim et al., 2024; Krishna et al., 2017). To improve video captioning, PLLaVa (Xu et al., 2024) builds on top of LLaVa and introduced a parameter-free pooling strategy to enhance the caption quality. Video-LLaVA (Lin et al., 2023a) achieves state-of-the-art performance on several benchmarks by conducting joint training on images and videos, thereby learning a unified visual representation. Video-LLaMA (Zhang et al., 2023) incorporates both video and audio into LLMs by introducing two Q-formers to extract features. Vid2seq (Yang et al., 2023) conducts large-scale pre-training with narrated videos for dense video captioning. Meanwhile, MV-GPT (Seo et al., 2022) employs an automated speech recognition (ASR) model to provide additional labeling for the videos.

**LLM-based Summarization.** Recently many works have found that it is efficient to summarize useful information using LLMs. For example, LLaDA (Li et al., 2024) can provide users with helpful instructions based on the user request and corresponding traffic rules in the desired location. OpenAI team finds re-captioning (Betker et al., 2023) via LLMs can be very helpful.

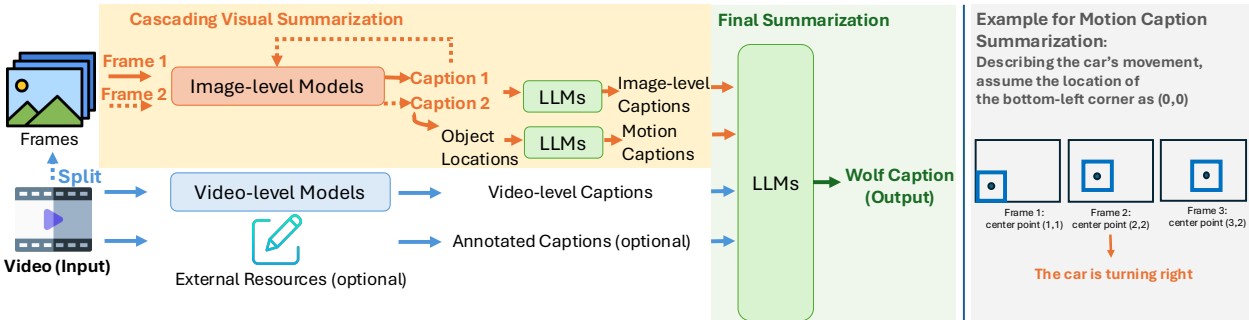

Figure 1: Overview of proposed Wolf framework. Wolf utilizes both image-level and video-level models to generate diverse and detailed captions, which are then summarized for cross-checking. On the right side, we also provide an example of how we obtain motion captions based on object locations extracted from image captions.

## 3 Wolf Framework

We propose Wolf, which is an automated dense captioning summarization framework that adopts a mixture of experts approach to generate long, accurate, and detailed captions for videos. Figure 1 provides an overview of our framework. In this paper, we use CogAgent (Hong et al., 2024), GPT-4V (Mao et al., 2023a) to generating image-level captions, and use VILA-1.5-7B (Lin et al., 2023c), Gemini-Pro-1.5 (Team et al., 2023) to generate video captions.

**Cascading Visual Summarization.** As image-level models (image-based VLMs) have been pre-trained with a larger amount of data than video-level models (video-based VLMs), we first use image-based VLMs to generate captions. We design a cascading visual summarizing program to obtain video captions from image-level models. As illustrated in Figure 1, we first split the video into sequential images, sampling two key-frames every second. We **start by** feeding Image 1 into the Image-level Model to obtain Caption 1, where we require the model to generate detailed scene-level information and object locations. Given the temporal correlation between key frames in a video, we then feed both Caption 1 and Image 2 into the model to generate Caption 2. By repeating this procedure, we generate captions for all sampled frames. Finally, we use GPT-4 to summarize the information from all captions with the prompt "*Summarize all the captions to describe the video with accurate temporal information*". We also extract the bounding box locations for each object in each frame, then feed them into LLMs to summarize the trajectory of the moving object. For example, in a driving video, a blue car is driving into the right lane, and the centers of the bounding boxes are (0,0), (1,1), (1,2). We provide the car's location to the LLM, and it outputs 'the blue car is driving to the right,' which we refer to as a '*Motion Caption*'.

**LLM-based Video Summarization.** Besides obtaining the captions from image-level models, we then summarize all captions into one. We use the prompt "*Please summarize on the visual and narrative elements of the video in detail from descriptions from Image Models (Image-level Caption and Motion Caption) and descriptions from Video Models (Video-level Caption)*". Optionally, we can also add the Annotated Caption to the summarization. Based on this simple scheme, Wolf can capture a rich variety of details of the video and reduce hallucinations (in Figure 2). We assume this is because Wolf can compare the captions and reduce redundant and hallucinated information. After obtaining the descriptions from the image-level and video-level models, we next apply the prompt "*Please describe the visual and narrative elements of the video in detail, particularly the motion behavior*".

## 4 Benchmarking Video Captioning

To showcase the effectiveness of Wolf, we constructed four distinct datasets (please check the examples in Figure 2. These include two autonomous driving video captioning datasets based on the open-sourced NuScenes (Caesar et al., 2019) dataset (Creative Commons Attribution-NonCommercial-ShareAlike 4.0

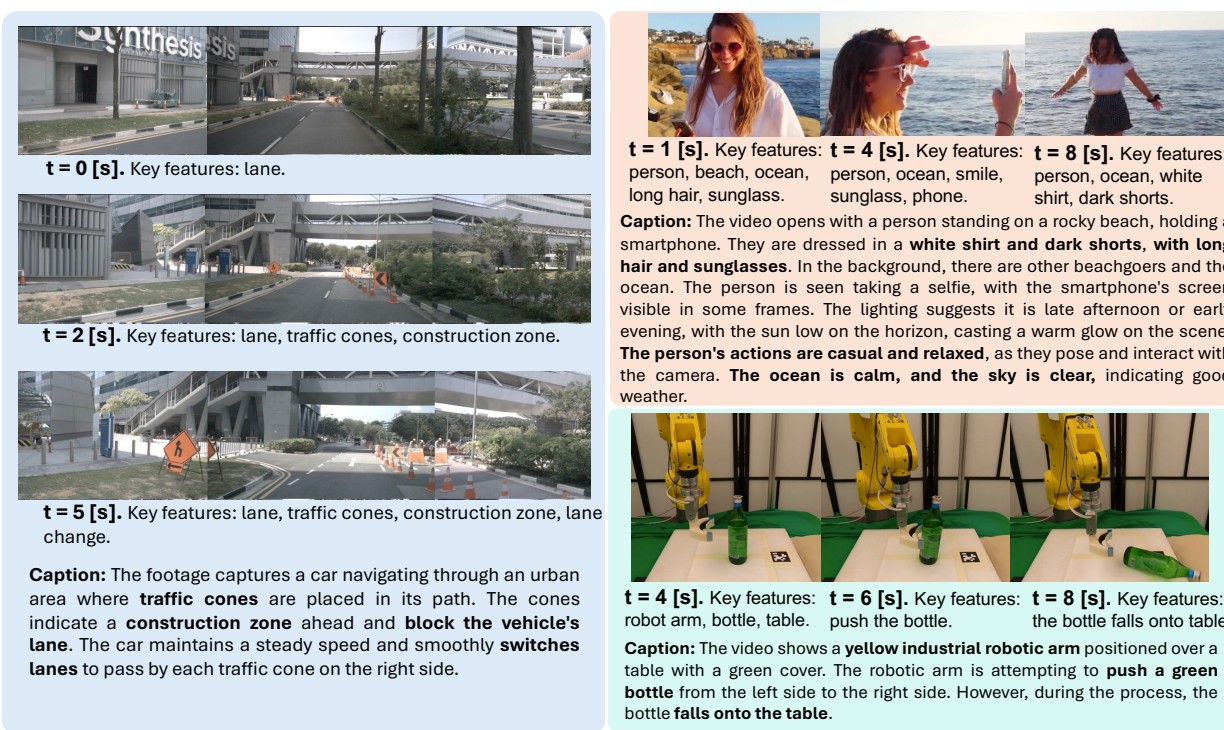

Figure 2: Wolf Dataset examples. We display the videos and corresponding human-annotated captions of autonomous driving (*Left*), Pexels (*Top-Right*), and Robot learning video dataset (*Bottom-Right*), totaling 25.7 hours. Our Wolf dataset is fully manually annotated to ensure a robust evaluation for the community. We present our dataset's statistics in Table 1. We will keep updating and expanding the dataset.

International Public License), a general daily video captioning dataset from Pexels [1], and a robot manipulation video captioning dataset from an open-source robot learning dataset (Padalkar et al., 2023). These benchmark datasets are tailored to assess the caption model's scene comprehension and its behavior understanding capabilities, both of which are vital for auto-labeling in embodied AI tasks. All captions were generated using a combination of ground truth information, rule-based heuristics, human labeling, and rewriting.

## 4.1 Wolf Dataset Curation

### 4.1.1 Autonomous Driving Dataset

High-quality captions of driving videos are crucial not only for training video generation models but also for training VLMs to interpret the dynamic traffic environment. The NuScenes dataset is a large-scale collection of driving videos designed to accelerate autonomous driving research. It features 1,000 annotated scenes from Boston and Singapore. Each scene consists of a 20-second driving video clip that provides an ego-centric view from the ego vehicle. We split each scene into 5-second segments and provide the corresponding captions. Our captions emphasize the high-level driving behavior of the ego vehicle to stress-test the scene understanding ability and the behavior understanding ability of a captioning model. Our dataset contains 500 intensely interactive video-caption pairs (≈0.7 hours) in which the ego vehicle is involved in intense interactions with its surrounding traffic agents (such as navigating around construction zones and overtaking static obstacles) and 4785 normal driving scene video-caption pairs (≈6 hours). Our caption generation process consists of three steps: i) agent-level motion annotation, ii) ego-centric interaction annotation, and iii) information aggregation via LLM.

**Step 1: agent-level motion annotation.** The NuScenes dataset provides full annotations of traffic elements in each scene, including 3D bounding boxes, element categories, and semantic map information. Similar

---

[1] https://www.pexels.com/

Figure 3: Illustration of homotopy types of different relative motions between a pair of vehicles.

to DriveVLM (Tian et al., 2024), we utilize this ground truth data along with lane topology information (Naumann et al., 2023) to generate text descriptions of both speed and angular motion characteristics for the ego vehicle and other traffic participants within a video clip. Specifically, we classify agent actions into 11 categories, including Stopping, Accelerating, Decelerating, Lane Changes, Turns, and more, based on their observed movements and behaviors.

**Step 2: egocentric interaction annotation**. Beyond each agent's dynamics information, we also aim to capture the ego vehicle's interactions with other traffic participants (e.g., crossing pedestrians, blocking traffic cones) depicted in the video clip. To efficiently describe interactions, we use two categorical modes: the lane relationship (*agent-ego lane mode*) and relative motion (*homotopy*) between a traffic participant and the ego vehicle (Chen et al., 2023c). At each time step $t$, the agent-ego lane mode encodes the topological relationship between the ego vehicle's current lane and the traffic agent's lane. The categories include *LEFT*, *RIGHT*, *AHEAD*, *BEHIND*, and *NOTON*, where *NOTON* indicates that the traffic agent is on a lane that cannot directly reach the ego vehicle's lane. To compute the agent-ego lane mode, we follow (Chen et al., 2023c) by identifying each agent's lane and using a lane topology map for annotation. Homotopy describes the relative motion between agents in a video and is categorized as: [*S*, *CW*, *CCW*] (*static, clockwise, counterclockwise*), as shown in Figure 3.

**Step 3: information aggregation.** By combining agent-ego lane mode, homotopy, traffic agents' ground truth dynamics, and scene context (e.g., the ego vehicle is near an intersection), we can apply heuristics to annotate interaction descriptions. For example, in a video clip, a static object's agent-ego lane mode changes from *AHEAD*, to *LEFT*, to *BEHIND*, and the ego vehicle's first performs *RIGHT-LANE-CHANGE*, *KEEP-LANE*, then *LEFT-LANE-CHANGE*, indicating the ego vehicle overtakes that object from the ego vehicle's left side. We identified six interaction categories from the NuScenes dataset: 1) bypass blocking traffic cones to navigate around construction zone; 2) yield to crossing pedestrians; 3) yield to incoming vehicles; 4) overtake traffic agents via straddling the lane dividers; 5) overtake traffic agent via lane-change; 6) other non-intensive interactions. With both agent-level motion annotations and ego-centric interaction annotations, we employ an LLM to aggregate this information and generate a human-like scene description. While any off-the-shelf LLM could be used for this task, we opted for the GPT-3.5 model. Additionally, we experimented with the llama 3 model and observed similar performance.

| Task Type | Source | Size | Annotation Type |
|---|---|---|---|
| Normal Driving Scenes | Nuscenes | 4,785 | Manually |
| Challenging Driving Scenes | Nuscenes | 500 | Manually |
| General Daily Scenes | Pexels | 473 | Manually |
| Robot Manipulation | UCB | 100 | Manually |

Table 1: Statistics of the Wolf dataset.

### 4.1.2 Robot Manipulation Dataset

In addition to the driving environment, we collect 100 robot manipulation videos (each has a length ranging from 5 seconds to 1 minute) from Padalkar et al. (2023) that demonstrate complex robot manipulations (e.g., pick and place, push, ect.) in various environments, including kitchen, office, lab, and open world. We

manually caption each video. The captions focus on the description of the scene and the interaction between the robot and the objects.

### 4.1.3   Pexels Dataset

To evaluate caption models in general daily environments, we further collect high quality (360p to 1080p) videos from Pexels. It consists of 473 high-quality videos sourced globally, where each video has a length varying between 10 seconds and 2 minutes and the content includes 15 popular categories (details in Appendix). This diversity not only adds depth to our dataset but also provides a wide range of scenarios and contexts for our analysis.

## 4.2   Wolf Evaluation Metric

### 4.2.1   CapScore: Evaluating Captions with LLMs

Video captioning has been an ill-posed problem since there is no metric to evaluate the quality of captions and the alignment between the video and the caption. Inspired by BERTScore (Zhang et al., 2019), CLIPScore (Hessel et al., 2021) and the stability of LLMs on evaluation (Chan et al., 2023; Lin et al., 2025; 2023d), we introduce **CapScore** (Captioning Score), a quantitative metric to use LLMs to evaluate the similarity between predicted and human-annotated (ground truth) captions. We tried both GPT-4 (model="gpt-4") and Llama 3.2 (Dubey et al., 2024) as our LLM to summarize the captions. We noticed that GPT-4 can always obtain stable results over 3 runs. However, for Llama 3.2, the results varied over different runs. We tried to lower the temperature (from 0.9 to 0.5) to make the inference stable, however, we noticed that the scores are not consistent with human evaluation. Therefore we select GPT-4 as our LLM to conduct the experiments. Assume we have 6 captions, we feed all the captions into GPT-4 and add the prompt "*Can you give a score (two decimal places) from 0 to 1 for captions 1, 2, 3, 4 and 5, indicating which one is closer to the ground truth caption (metric 1) and which contains fewer hallucinations and less misalignment (metric 2)? Please output only the scores of each metric separated only by a semicolon. For each metric, please output only the scores of captions 1, 2, 3, 4 and 5 separated by commas, in order—no text in the output.* ". We ask GPT-4 to output two scores: caption similarity and caption quality.

We set the range [0,1] to align with several widely used NLP metrics, such as BLEU (Papineni et al., 2002), ROUGE (Lin, 2004), and BERTScore (Zhang et al., 2019). To address the potential concern, we followed the same settings as Table 1 and used the range [0,5] to calculate CapScore. The trend remains precisely the same, with Wolf achieving scores of 3.61 for similarity and 3.70 for quality - almost five times the values shown in Table 1, demonstrating CapScore's stability and robustness regardless of the range.

**Caption Similarity.** Caption similarity is based on how well each caption aligns with the ground truth description on a scale from 0 to 1, considering the key criteria mentioned. GPT-4 lists the requirements that affect the score: this metric measures how similar each caption is to the ground truth caption. The evaluation focuses on the content and context described in the captions, assessing whether they capture the main themes and details of the ground truth.

**Caption Quality.** Caption quality evaluates whether the caption contains reduced hallucination and mistakes compared to the ground truth captions on a scale from 0 to 1. GPT-4 lists the criteria that affect the score: this metric evaluates the accuracy and relevance of each caption, identifying any extraneous or incorrect details (hallucinations). Captions with fewer hallucinations and better alignment receive higher scores.

### 4.2.2   Human-Evaluation Score and CapScore

Through our experiments, we find that GPT-4 is very robust for calculating the scores. We have run the experiments for 1-3 times, the results appear to be stable and less than 0.05 changes. To alleviate concerns related to human alignment and correlation, we randomly selected 10 users to evaluate our set of 100 robotics videos, as detailed in Table 1 of the paper. The evaluators were presented with the videos, the generated captions, and the corresponding ground truth captions. We asked them to assign human-evaluation scores based on the CapScore standard, with the following prompt: "*After reviewing the video and all the captions, please assign the caption similarity and caption quality score (floating point values) from 0 to 1 for different*

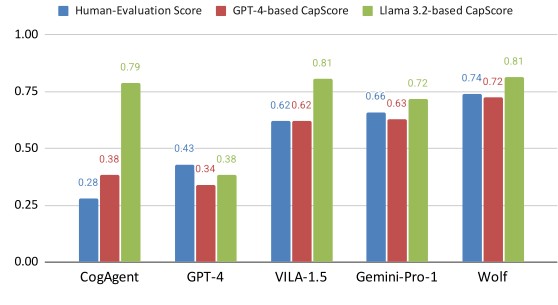

(a) Comparison on Caption Similarity.

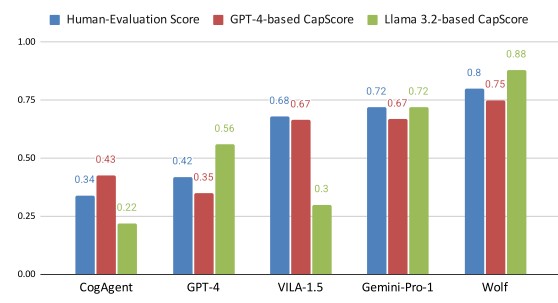

(b) Comparison on Caption Quality.

Figure 4: Comparisons on Human-Evaluation Score and Llama 3.2-based CapScore and GPT-4-based CapScore (proposed).

*captions, indicating which caption is closest to the ground truth (caption similarity) and which one has fewer hallucinations and less misalignment (caption quality)."* We show the results in Figure 4. We discover **CapScore** is stable and aligns with trends of human evaluation. We calculated the Pearson correlation coefficient, obtaining 0.93 and 0.95 for caption similarity and quality, which further indicate a strong positive correlation between human evaluation and CapScore. Beyond that, we also conduct experiments comparing CapScore with other widely used image captioning evaluation metrics, as is shown in Appendix (Sec A.5). We observe that CapScore aligns with trends observed in other metrics but highlights a larger performance gap between models, suggesting it serves as a more effective evaluation metric.

### 4.2.3 Benchmarking Video Captioning

To our best knowledge, no standard evaluation benchmarks have been established for video understanding and captioning. To accelerate the advancement of this field, we have developed the first leaderboard for video captioning. As LLM evaluation has become increasingly popular (Chiang et al., 2024), we realized the lack of a standard platform to evaluate VLM's performance on video understanding. We assume this is due to the difficulty of collecting ground truth captions that accurately align with videos. We will release the initial version of our captioning leaderboard upon publication.

## 5 Experiments

### 5.1 Experimental Setup

**Data Setup.** We use four sets of data to evaluate the validity of Wolf: 1) 500 Nuscences Interactive Videos; 2) 4,785 Nuscences Normal Videos; 3) 473 general videos and 4) 100 robotics videos. We extract 2 frames per second for autonomous driving videos. For robotics videos, we extract 1 frame per second. For short videos that sample less frames, we will increase `fps` to capture more details.

**Comparison Setup.** We use our proposed CapScore to evaluate the similarity between predicted and ground truth captions. CogAgent and GPT-4V are image-level methods, so we upload sequential frames into the model to obtain the output. VILA-1.5-7B and Gemini-Pro 1.5 are video-based, so we directly feed a video into the model. As for the prompt for each captioning model, we use "*elaborate on the visual and narrative elements of the video in detail, particularly the motion behavior*". We compare with four widely-used image-level and video-level captioning Vision-Language Models (VLMs) CogAgent (Hong et al., 2024), GPT-4V (Achiam et al., 2023), VILA-1.5 (Lin et al., 2023c) and Gemini-Pro-1.5 (Team et al., 2023). As for CogAgent, we feed the middle frame of the video into the model to obtain the captions. As for GPT-4V, we uniformly sample 16 frames from a video and feed the sequential images into the model to obtain captions. As for VILA-1.5-7B and Gemini-Pro-1.5, we feed the video into the model to obtain the captions.

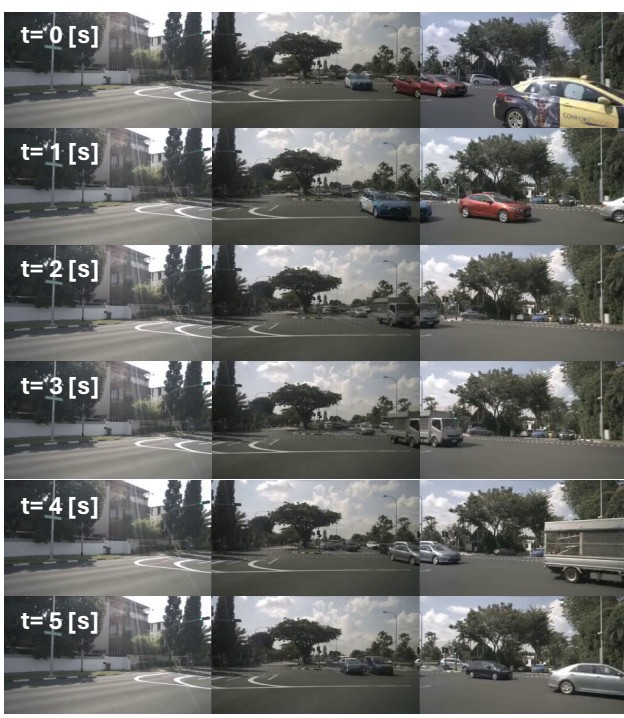

t=0 [s]
t=1 [s]
t=2 [s]
t=3 [s]
t=4 [s]
t=5 [s]

**Wolf** 🐺 : The video presents a series of scenes depicting city streets bustling with various vehicles such as cars, trucks, motorcycles, and a person riding a bicycle. The scenes are often divided into panels, each showcasing a different vehicle in motion, including a yellow taxi, a blue car, a red car, a white truck, and a silver car. The streets are marked with white lines and are surrounded by green trees and buildings, contributing to an urban landscape setting. The video captures the typical flow of traffic through a four-way intersection, with vehicles **moving in different directions**. The vehicles are shown in various positions in the lanes, implying movement through the area. The sequence likely shows vehicles **accelerating** and **decelerating** as they approach and leave the intersection in response to traffic signals or the flow of other vehicles. The video is a wide-angle shot of the street intersection, split into three sections, providing a panoramic view. The left section shows the road the viewer is on, the center focuses on the intersection itself, and the right section provides a view of the crossing traffic. The lighting suggests a sunny day, with a bright spot from the sun on the left section and more even lighting on the right. The video captures the typical flow of traffic at an intersection, with vehicles approaching, some **waiting their turn while others proceed through the intersection when it's clear**. The cars driving on the right side of the road suggest the location might be a country where right-hand drive is the standard. The absence of sound enhances the focus on the visual information and allows the viewer to interpret the scene without auditory cues. Overall, the video provides a snapshot of everyday life at a **busy intersection**, highlighting the **complexities of navigating a traffic intersection**. The wide-angle perspective and focus on movement create a sense of dynamism.

Figure 5: Wolf example for driving that focus on interactive operations. Wolf captions discusses the motion behavior in details and serves as a good reference for autonomous driving. Note: Please refer to the Appendix for our caption comparison with other state-of-the-art methods.

## 5.2 Qualitative Results

To illustrate enhanced captioning ability by Wolf, we show the qualitative results in Figure 5 (please check details in Appendix). We noticed that although GPT-4V is good at recognizing the scenes, capturing temporal information in a video is not ideal. Gemini-Pro-1.5 can capture video information such as "waiting their turn while others proceed through the intersection when it's clear", but it fails to describe the detailed motions. In comparison to these two state-of-the-art approaches, we observed that Wolf not only captures the motion described in Gemini-Pro-1.5 but also successfully captures "vehicles moving in different directions" and "vehicles accelerating and decelerating as they approach and leave the intersection in response to traffic signals or the flow of other vehicles".

## 5.3 Quantitative Results

We compare Wolf with various state-of-the-art captioning models and display the results on 4 datasets in Table 3 and 2. In the default setting, Wolf uses CogAgent, GPT-4V, VILA-1.5-7B, and Gemini-Pro-1.5 as

| Method | Caption Similarity ↑ | | | Caption Quality (eg. reduced hallucination) ↑ | | |
|---|---|---|---|---|---|---|
| | Nuscenes | Pexels | Robotics | Nuscenes | Pexels | Robotics |
| CogAgent (Hong et al., 2024) | 0.18 | 0.68 | 0.38 | 0.24 | 0.72 | 0.43 |
| GPT-4V (Achiam et al., 2023) | 0.31 | 0.72 | 0.34 | 0.36 | 0.75 | 0.35 |
| VILA-1.5-7B (Lin et al., 2023c) | 0.21 | 0.85 | 0.62 | 0.25 | 0.86 | 0.67 |
| Gemini-Pro-1.5 (Team et al., 2023) | 0.42 | 0.87 | 0.63 | 0.45 | 0.87 | 0.67 |
| **Wolf** | **0.55** | **0.88** | **0.72** | **0.56** | **0.89** | **0.75** |

Table 2: Comparison on 500 highly interactive (difficulty and challenging) Nuscenes videos, 473 Pexels videos and 100 robotics videos. Our Wolf exhibits better performance than both open- and closed-source models.

| Method | Caption Similarity ↑ | Caption Quality ↑ |
|---|---|---|
| CogAgent (Hong et al., 2024) | 0.27 | 0.30 |
| VILA-1.5 (Lin et al., 2023c) | 0.35 | 0.39 |
| **Wolf** (based on VILA-1.5-7B) | **0.56** | **0.60** |

Table 3: Comparison on 4,785 normal Nuscenes videos. The quality of Wolf is consistently better.

| VILA-1.5-7B | Caption Similarity ↑ | Caption Quality ↑ |
|---|---|---|
| Default | 0.21 | 0.25 |
| Fine-tuned with Wolf annotation | **0.36** | **0.37** |

Table 4: Comparison on 500 highly interactive Nuscenes videos VILA-1.5 and fine-tuned VILA-1.5 with Wolf captions.

**Video-level models.** Due to the running cost, we use Wolf (based on VILA-1.5) on the Nuscenes Normal dataset, which only uses CogAgent and VILA-1.5-7B. We notice that existing image-level models fail to capture the temporal information in detail. Video-level models perform better, while Wolf can achieve the best results compared to all state-of-the-art captioning models. We also observe that all VLMs perform reasonably well on general daily scenes; however, they perform quite poorly on robotics and driving datasets. We assume this is due to the lack of training data for each individual model. Therefore, Wolf can effectively address this issue by distilling and summarizing knowledge from different models.

## 5.4 Finetuning VLMs with Wolf Captions

### 5.4.1 Comparison on Wolf Dataset

To further verify the effectiveness of Wolf, we finetune VILA-1.5-7B based on Wolf's captions on 4,785 normal Nuscenes videos and evaluate it on 500 highly interactive Nuscenes videos, which have much more difficult captions and complex scenarios. We follow the original VILA's training setup and launch supervised-finetuning with Wolf generated video-caption pairs for one epoch. The training is performed on 8xA100 GPUs with batch size 8. We set the learning rate to $10^{-4}$ with warmup strategy. No weight decay is applied. We demonstrate the results in Table 4, corresponding to Table 2. We observe that finetuning with Wolf boosts the model performance to 71.4% on caption similarity and 48.0% on caption quality, which outperforms GPT-4V and approaches Gemini-Pro-1.5. This suggests that Wolf captions can be easily applied to push VLMs' performance to a higher level.

### 5.4.2 Comparison on Other Benchmark Datasets

To scalable measure the quality of captions, we compare the VILA-1.5-13B trained w/ Wolf captions and w/o Wolf captions to study the effectiveness. We benchmark the Wolf-finetuned models on two widely used video datasets ActivityNet (Caba Heilbron et al., 2015) and MSRVTT (Xu et al., 2016) and display the results in Table 5, the improved performance effectively demonstrates the efficiency of Wolf.

| VILA-1.5-13B | ActivityNet | MSRVTT |
|---|---|---|
| Default | 54.7 | 60.2 |
| Fine-tuned with Wolf annotation | **55.2** | **60.9** |

Table 5: QA Accuracy comparison of the fine-Tuned Model on Activity and MSRVTT datasets.

| Method | Caption Similarity ↑ | Caption Quality ↑ |
|---|---|---|
| CogAgent | 0.18 | 0.24 |
| Wolf CogAgent part (Cascading Visual Summarization) | **0.26** | **0.32** |
| Wolf video part (VILA-1.5-7B+Gemini-Pro-1.5+GPT-4V) | 0.40 | 0.42 |
| Wolf (based on VILA-1.5-7B) | 0.35 | 0.37 |
| Wolf (based on VILA-1.5-7B+Gemini-Pro-1.5) | 0.48 | 0.49 |
| Wolf (based on VILA-1.5-7B+Gemini-Pro-1.5+GPT-4V) | **0.55** | **0.56** |

Table 6: Ablation study on 500 highly interactive Nuscenes videos. Note: The first row shows the results using *only image-level models*, the second row shows the results using *only video-level models*, and the last row shows the results using *both image-level models (CogAgent part) and various video-level models*.

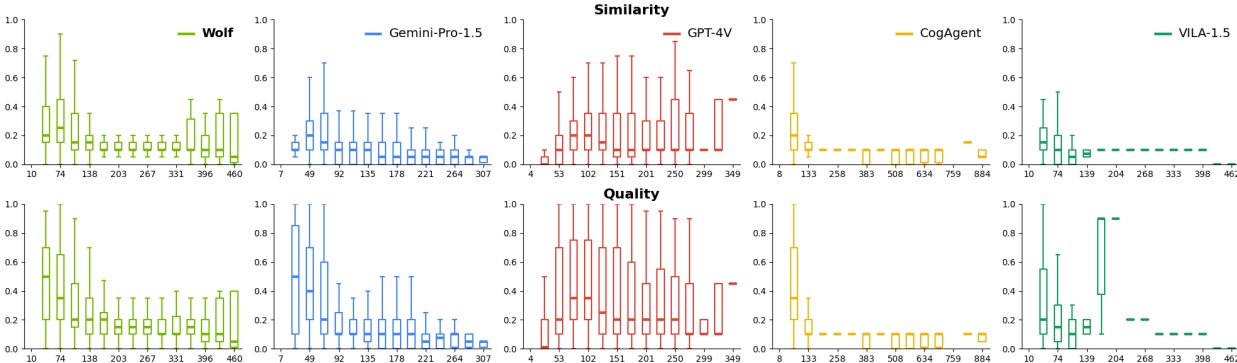

Figure 6: CapScore Caption Similarity and Caption Quality evaluated under varying caption length.

## 5.5 Ablation Study on Video-level Model Selection

To further evaluate how various video-level models affect the performance, we conduct an ablation study on the components of the models in Table 6. We first compare the caption from the middle frame of CogAgent with Wolf Caption based on the visual cascading summarization approach (only using CogAgent). The visual cascading summarization procedure could largely improve the video understanding quality from an image-level model such as CogAgent. Then, we conduct an ablation using only the video-level models. Finally, we compare Wolf with various combinations of video captions. We notice that Wolf consistensly shows better CapScore as its dense framework reduces hallucination and incorporate video details from different models.

## 5.6 Ablation Study on Token Efficiency

It is well-known that the LLMs finetuned with RLHF favor longer response (Singhal et al., 2023), a phenomenon referred to as verbosity issue. To better assess the efficiency of the captions, we performed additional evaluation using the CapScore judge. Specifically, we separate each caption result into sentences, then incrementally use more sentences to form shortened captions, starting from only using the first sentence, to the whole original caption. These shortened captions are scored via CapScore, and we plot the score against the number of tokens used. We show the results in Figure 6.

From the result, we observe that for the better performing models (Wolf, Gemini-Pro-1.5 and GPT-4V) the similarity scores grow with token length when caption lengths are short, but quickly plateau or even drop as the caption lengths get too long. The caption quality score demonstrates quite diverse patterns from different models. GPT-4V maintains a relatively consistent quality score while Gemini-Pro-1.5 and Wolf display better quality when the caption length is short.

# 6 Discussion and Future Works

**Limitations and Optimization.** Wolf is still significantly more cost-effective for autolabeling and captioning than procuring human labels. However, there is an efficiency concern when using an ensemble method. This must be handled with care to ensure that GPU resources are used effectively to mitigate any throughput degradation compared to using single models, even though Wolf offers a significant improvement in caption quality. Modern GPUs are based on a massively parallel pipeline, and our goal is to saturate this pipeline with meaningful work. We consider three primary areas for optimization to make Wolf a unified and efficient framework: Low-Hanging Fruit, Batched Inference, and Model Quantization. For example, we reduce the size of the model weights for model quantization. Recent works (Lin et al., 2023b; Dettmers et al., 2024; Ma et al., 2024) have noted that LLMs and VLMs can produce highly accurate results even when their weights are quantized to low bit depths. Therefore, we quantize all constituent models used in Wolf to 4 bits to further improve efficiency. This has two benefits. First, it reduces the bandwidth required for computation. These algorithms work by packing two 4-bit numbers into a single 8-bit type, so when moving data on the GPU, only half the number of bits need to be moved. Since all currently released GPUs support native instructions on 8-bit floating point numbers, the two 4-bit numbers are extracted and expanded by each kernel. In other words, two computations can be performed for every move operation. Next-generation GPUs will natively support 4-bit data types, and we expect further efficiency improvements from having dedicated 4-bit multiply and add instructions. Second, it synergizes with batched inference since the model weights, which are traditionally 16-bit, now only require one quarter of the GPU memory they would ordinarily use. This allows us to fit larger batch sizes on each GPU and process more videos in parallel. Please check our Appendix for details.

**Safety Considerations.** As an ensemble of captioners, Wolf mitigates the possibility of missing out on crucial information in the captions and rectifying any hallucinations that do not agree with the output of most models, which is a fundamental pillar for developing safe autonomous systems, as specified in the functional safety standard ISO 26262 (ROHM). Beyond the benefits of Wolf, there are still various open questions pertaining to safety of VLM captioners in deployment which we aim to explore more in future: (i) We need to align the captions with the task at hand; e.g., in a driving scenario, a detailed description of the foliage around the road, even if correct, is irrelevant and can potentially act as distractor for the decision maker. (ii) Complementary to the first point, we need to *measure* how well a caption aligns with the task at hand and develop an advanced version of CapScore. (iii) Finally, we need an approach to quantify the confidence we have in the captions by leveraging techniques from learning theory, such as conformal prediction (Shafer & Vovk, 2008). Most prior work in this direction assumes an MCQ-styled outputs or those where a unique correct answer exists (Ren et al., 2023; 2024), but these approaches do not translate to free-form text descriptions.

# 7 Conclusion

In this work, we propose Wolf, a captioning framework designed to automatically and accurately annotate any video, with significant improvements in data alignment. We find out that adopting a mixture of captioning models and summarization can largely boost the quality of the captions. This enables obtaining long, detailed, and accurate video captioning. We will also establish a comprehensive library that includes various types of videos with high-quality captions, regional information such as 2D and 3D bounding boxes and depth, as well as multiple object motions and interactions.

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

## A  Contributions

We would like to list Wolf Contributions:

**1) Framework and Evaluation Metric.** We designed a novel world summarization framework, **Wolf**, for video captioning and introduced an LLM-based metric, **CapScore**, to evaluate the quality of captions. The results show that our method significantly improves CapScore.

**2) Datasets and Benchmark.** We introduce the Wolf benchmark (leaderboard) and four human-annotated benchmark datasets. These datasets include autonomous driving, general scenes from Pexels, robotics videos, and human-annotated captions, collectively referred to as the **Wolf Dataset**.

**3) Intended Uses.** We believe Wolf can serve as one of the best practices (auto-labeling tool) for creating and curating paired datasets and benchmarks.

**4) Hosting, licensing, and maintenance plan.** The code, data, and leaderboard will be open-sourced and maintained. Continuous efforts will be made to refine the Wolf Dataset, Wolf codebase, and CapScore. We hope that Wolf will raise awareness about the quality of video captioning, set a standard for the field, and boost community development.

## B  Pexel Dataset Categories

We categorize videos from pexel into the following types: Travel & Events, Sports, Education, Pets & Animals, People & Blogs, Nonprofits & Activism, News & Politics, Music, Science & Technology, Comedy, Entertainment, Film & Animation, Gaming, Robotics, How to Styles.

## C  Qualitative Caption Comparison on Interactive Nuscenes Driving Videos

We display the details of Figure 4 of the paper (Wolf example for driving videos that focus on interactive operations) in Figure 7.

## D  Wolf Efficiency Optimization

We consider three primary areas: **Low-Hanging Fruit**, **Batched Inference**, and **Model Quantization** as optimizations which make Wolf a unified and efficient framework. Using the optimizations detailed in this section we were able to increase the speed of CogVLM by a factor of approximately 10x (450s/video to 41s/video), VILA throughput was similarly improved to only about 3s per video.

**Low-Hanging Fruit.** These are primarily systems concerns and work arounds for simplistically written APIs. For example, the off-the-shelf CogVLM (Hong et al., 2024) and VILA (Lin et al., 2023c) supporting code is heavily based on loading PIL images to present to a huggingface pipeline (Wolf et al., 2019). In the naive pipeline, videos would need to be decoded and then converted to PIL images before input to the respective pipelines, which in turn convert them to GPU PyTorch (Ansel et al., 2024) tensors. This is extremely inefficient. Instead, we can leverage the hardware video decoder present in modern GPUs to decode the videos directly to GPU tensors and rewrite the preprocessing pipelines to operate on these tensors directly. This has the additional benefit of shifting preprocessing transform work from CPU to GPU.

**Batched Inference.** Simplifying Wolf into the simplest terms, we are essentially performing repeated neural network inference. Surprisingly, most VLM supporting code is designed to run inference on only a single example at a time. However, just as in other deep-learning problems, there fundamentally no reason why we cannot processes multiple videos at a single time in batches. This step is crucial to maximizing the use of GPU resources. Processing a single example may only use as little as 25% of a modern datacenter GPU which would either increase the time to process a dataset or the number of GPUs required to complete a task in a fixed time budget. We can reimplement more of the supporting code to enable processing batches of as many videos as will fit in GPU memory at a single time yielding a linear speedup in processing. For example, if we can fit batches of 4 in GPU memory we observe a speedup of 4x over processing single examples.

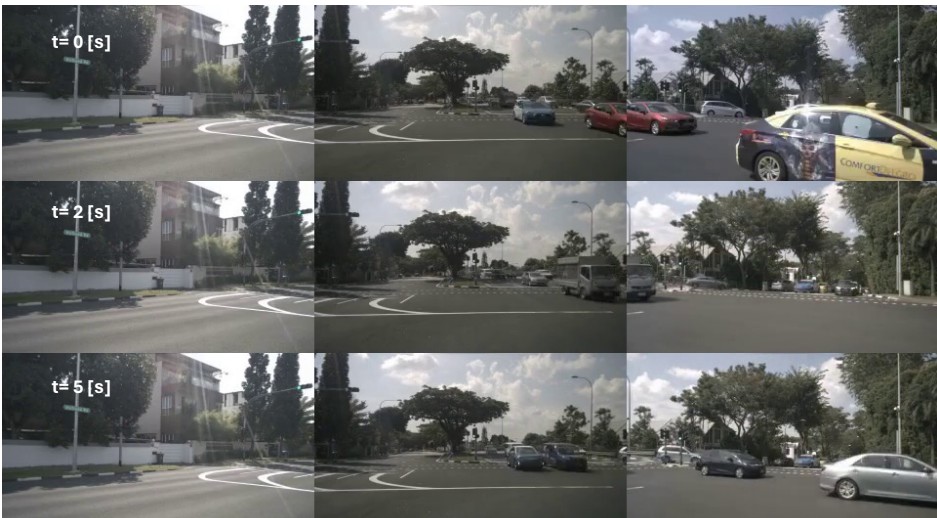

**CogAgent**: The picture shows a three-panel scene of a city street. In the first panel, a white truck is driving on the road, and there are several cars and motorcycles on the street. In the second panel, the white truck is still driving on the road, and there are several cars and motorcycles on the street. In the third panel, the white truck is driving on the road, and there are several cars and motorcycles on the street.

**VILA-1.5**: The video presents a sequence of images showing a street view with a pedestrian crossing. The images feature a white truck parked on the left side of the frame, a pedestrian crossing the street, and a pedestrian crossing sign visible on the right side. The sky is clear with few clouds, and the buildings in the background have a modern architectural style. The street is lined with trees and there are multiple cars parked on the right side of the street. The pedestrian is wearing dark clothing and is seen walking from one side of the street to the other.

**GPT-4**: It appears that you've provided a series of still images which seem to be sequential frames from a video that captures a scene at a traffic intersection. The visual elements include the following: a road with multiple lanes, white lane markings, and a variety of vehicles, including cars and a truck, navigating the intersection. The vehicles are captured in various positions in the lanes, implying movement through the area. There appears to be a tree-lined median or central island, and trees and buildings can be seen in the background, contributing to an urban landscape setting. The motion behavior depicted across these frames suggests the typical flow of traffic through an intersection. Vehicles are moving in a coordinated manner, adhering to lane directions and traffic signals that are implied rather than visible in the images. The sequence likely shows vehicles accelerating and decelerating as they approach and leave the intersection in response to traffic signals or the flow of other vehicles. The spatial positioning of the vehicles changes progressively, implying a real-time capture of their movements. If the sequence was put together as a continuous video, you would see vehicles entering the frame from one side and exiting on the other, with varying speeds and some vehicles possibly turning, illustrating the dynamic and fluid nature of traffic movement within a regulated urban environment.

**Gemini-Pro-1.5**: The video is a wide-angle shot of a street intersection. The image is split into three sections, providing a panoramic view. Here's a breakdown: Visual Elements: Perspective: The three-part image creates a wide-angle perspective, simulating the view a driver might have at the intersection. The left section shows the road the (assumed) viewer is on, the center focuses on the intersection itself, and the right section provides a view of the crossing traffic. Color Palette: The video appears to be in color, but the colors are somewhat muted, giving it a realistic but slightly desaturated feel. Lighting: The lighting suggests a sunny day. The left section has a bright spot from the sun, possibly late afternoon, while the right section is more evenly lit. Static Elements: Elements like trees, buildings, lampposts, and road markings provide context and a sense of place. Dynamic Elements: Vehicles, including cars and a truck, move through the intersection, creating a dynamic scene. Narrative Elements/Motion Behavior: Traffic Flow: The video captures the typical flow of traffic at an intersection. Vehicles approach, some wait their turn while others proceed through the intersection when it's clear. Right-Hand Drive: The cars driving on the right side of the road suggest the location might be a country where right-hand drive is the standard. Implied Viewer: While there is no visible camera or person, the perspective of the shot implies the viewer is positioned within a vehicle approaching the intersection. No Sound: The absence of sound enhances the focus on the visual information and allows the viewer to interpret the scene without auditory cues. Overall Impression: The video provides a snapshot of everyday life at a busy intersection. The wide-angle perspective and focus on movement create a sense of dynamism and highlight the complexities of navigating a traffic intersection.

**Wolf**: The video presents a series of scenes depicting city streets bustling with various vehicles such as cars, trucks, motorcycles, and a person riding a bicycle. The scenes are often divided into panels, each showcasing a different vehicle in motion, including a yellow taxi, a blue car, a red car, a white truck, and a silver car. The streets are marked with white lines and are surrounded by green trees and buildings, contributing to an urban landscape setting. The video captures the typical flow of traffic through a four-way intersection, with vehicles **moving in different directions**. The vehicles are shown in various positions in the lanes, implying movement through the area. The sequence likely shows vehicles **accelerating** and **decelerating** as they approach and leave the intersection in response to traffic signals or the flow of other vehicles. The video is a wide-angle shot of the street intersection, split into three sections, providing a panoramic view. The left section shows the road the viewer is on, the center focuses on the intersection itself, and the right section provides a view of the crossing traffic. The lighting suggests a sunny day, with a bright spot from the sun on the left section and more even lighting on the right. The video captures the typical flow of traffic at an intersection, with vehicles approaching, some **waiting their turn while others proceed through the intersection when it's clear**. The cars driving on the right side of the road suggest the location might be a country where right-hand drive is the standard. The absence of sound enhances the focus on the visual information and allows the viewer to interpret the scene without auditory cues. Overall, the video provides a snapshot of everyday life at a busy intersection, highlighting the complexities of navigating a traffic intersection. The wide-angle perspective and focus on movement create a sense of dynamism.

Figure 7: Comparison of CogAgent, VILA-1.5, GPT-4, Gemini-Pro-1.5, and Wolf on Interactive Nuscenes Driving Videos, Corresponding to Figure 4 of the Paper.

| Model | CLIP-Score ↑ | CapScore$_S$↑ | CapScore$_Q$↑ | N-avg↑ |
|---|---|---|---|---|
| MiniGPT-4 | 0.601 | 0.330 | 0.359 | 0.19 |
| InstructBLIP | 0.599 | 0.360 | 0.355 | 0.18 |
| LLaVA-1.5 | 0.601 | 0.385 | **0.450** | 0.67 |
| mPLUG-Owl2 | 0.597 | **0.397** | 0.400 | 0.49 |
| Qwen2-VL | **0.618** | 0.373 | 0.432 | **0.82** |

Table 7: Comparison on CapScore and CLIP-Score for text-image alignment. CapScore$_S$ represents CapScore Similarity; CapScore$_Q$ represents CapScore Quality (such as reduced hallucination); N-avg represents noun/verb average. We observe that CapScore aligns with trends observed in other metrics but highlights a larger performance gap between models, suggesting it serves as a more effective evaluation metric. Note: All scores are scaled to the range [0, 1].

**Model Quantization.** The final optimization we consider is to reduce the size of the model weights. Several recent works (Lin et al., 2023b; Dettmers et al., 2024; Ma et al., 2024) have noted that LLMs and VLMs can produce highly accurate results even when their weights are quantized to low bit-depths. Therefore, we quantize all constituent models used in Wolf to 4-bits to further improve efficiency. This has two benefits. First, it reduces the bandwidth required for computation. These algorithms work by packing two 4-bit numbers into a single 8-bit type, so when moving data on the GPU only half the number of bits need to be moved. Since all currently released GPUs support native instructions on 8-bit floating point numbers, the two 4-bit numbers are extracted and expanded by each kernel. In other words, two computations can be performed for every move operation. Next generation GPUs will natively support 4-bit datatypes and we expect further efficiency improvements from having dedicated 4-bit multiply and add instructions. Next, it synergizes with batched inference since the model weights, which are traditionally 16-bit, now only require one quarter of the GPU memory they would ordinarily use. This allows us to fit larger batch sizes on each GPU and process more videos in parallel.

# E  Comparing CapScore with Other Metrics

To verify the efficiency of CapScore, we compare CapScore with the two most widely used captioning scores: 'CLIP-Score' (Hessel et al., 2021) and 'Noun and verb coverage' (N-avg) (). Using CLIP, the *CLIPScore* between the image $I$ and all the generated captions is computed. *Recall@k* is calculated to determine whether the corresponding generated caption $y'$ appears within the top-$k$ most similar captions. N-avg assesses how well a caption $y'$ covers key objects (nouns) and actions (verbs) present in an image by comparing it to the groundtruth $y$.

Noun coverage is calculated as:

$$\text{Noun Coverage} = \frac{|N(y) \cap N(y')|}{|N(y')|} \tag{1}$$

where $N(y')$ is the set of all nouns in $y'$. Verb coverage is calculated for verbs likewise.

We evaluate various popular models on a wide-used image dataset COCO Karpathy test set (Karpathy & Fei-Fei, 2015): MiniGPT-4 (Chen et al., 2023a), InstructBLIP (Dai et al., 2023), LLaVA-1.5 (Liu et al., 2024), mPLUG-Owl2 (Ye et al., 2024) and Qwen2-VL (Wang et al., 2024). As is shown in Table 7, we observe that CapScore aligns with trends observed in other metrics but highlights a larger performance gap between models, suggesting it serves as a more effective evaluation metric.

