# OpenReview forum: "Wolf: Dense Video Captioning with a World Summarization Framework"
_TMLR — Accepted by TMLR_

### Review · Reviewer_HHVo · 2025-05-22

**Summary Of Contributions:**

This paper presents a comprehensive framework for generating and evaluating long-form video captions. The framework employs multiple specialized models to extract visual information from video frames and translate these features into natural language descriptions. LLMs are then utilized to synthesize and summarize the outputs from individual captioning models into coherent, comprehensive captions.

The authors contribute a challenging video captioning benchmark that include diverse domains, including autonomous driving scenarios, general everyday scenes, and robotics applications. Experimental results demonstrate that the proposed framework successfully generates high-quality captions for extended video sequences, addressing key challenges in long-form video understanding.

**Audience:**

Yes

**Claims And Evidence:**

Yes

**Requested Changes:**

The paper's writing quality requires substantial improvement, particularly in the introduction where several descriptions lack precision and clarity. The authors fail to adequately articulate the research motivation by clearly distinguishing their proposed long-form video captioning task from existing video captioning benchmarks. Additionally, statements such as "there is no established benchmark to measure captioning progress" are overly broad and inaccurate, as numerous video captioning benchmarks already exist in the literature.

The presented examples lack specificity regarding individual model outputs, making it difficult to assess each component's contribution. The notation "t=0 [s]....." is confusing and poorly explained. Furthermore, the origin and extraction methodology for the mentioned "key features" remain unclear, hindering reproducibility and understanding of the approach.

The paper lacks critical discussion of key design choices. The authors do not sufficiently justify why multi-model caption synthesis through LLMs is preferable to alternative approaches. Notably absent is an ablation study comparing their multi-stage approach against direct caption generation from visual representations using LLMs. This comparison would be essential to validate the proposed framework's necessity and effectiveness over simpler alternatives.

**Strengths And Weaknesses:**

The challenge of generating captions for long-form videos is a challenging topic, as isolated models typically struggle to maintain consistency and comprehensiveness across extended frames. The proposed approach effectively addresses this limitation by integrating outputs from multiple specialized models spanning both image and video domains, leveraging the complementary strengths of each modality.

The methodology is straightforward but effective, demonstrating that model integration can achieve superior performance without requiring complex architectural innovations. This simplicity makes the approach both interpretable and practically implementable.

The newly introduced benchmark exhibits high quality and fills an important gap in evaluation resources by incorporating diverse video types from autonomous driving, general scenes, and robotics domains, providing a more comprehensive assessment framework for long video captioning systems.

---

> ### Author Response · Authors · 2025-05-28
> **Response to Reviewer HHVo**
>
> We would like to thank Reviewer HHVo for taking the time to review our paper and provide valuable and insightful feedback, as well as for the quick review! We are pleased that the reviewer found our system to be "straightforward but effective" and appreciated our "our introduced benchmark fills an important gap in evaluation resources". We are happy to have the opportunity to address the reviewer’s questions and concerns.
>
> **Q1. Research motivation.**
>
> Thank you for your suggestion! We apologize for any confusion (including the reminder that the introduction may cause confusion) and will clarify this in the final version. We will revise the introduction in detail based on your comments. First, please allow us to explain the motivation behind our research. Existing video captioning benchmarks such as GOAL (2025), Video-MMMU (2025), and VidCapBench (2025) primarily focus on evaluating a model’s ability to perform question answering (Q&A) and grounded reasoning tasks. In contrast, Wolf emphasizes data curation, particularly the creation of paired captions for videos, which is crucial for various large model training tasks such as video understanding and video generation. There are two key differences in our approach: 1) We aim to accurately caption every critical detail across different scenarios. For instance, in autonomous driving, we want the model to precisely describe the car’s motion over time and in sequence. 2) Wolf evaluation is not limited to assessing reasoning capabilities. Instead, it aims to determine which generated captions best describe the video content. We observed that Q&A tasks do not effectively reveal whether a model’s caption is truly useful. Therefore, we introduced CapScore, which evaluates similarity and quality (e.g., reduced hallucination) by comparing generated captions to our ground-truth references. Regarding the statement that “there is no established benchmark to measure captioning progress,” we apologize for any confusion this may have caused, and we will revise this statement in the final version.
>
> **Q2.Each component's contribution and key features.**
>
> Thank you very much for your valuable suggestion. Since we are unable to include videos in the paper, we instead present frames at 0 seconds (t = 0 s), 1 second, and 2 seconds from the video. For example, on the left side of Figure 2, we show frames from an autonomous driving scenario, captured from the ego car’s perspective at these three time points. This approach is intended to illustrate how elements such as traffic cones and construction zones evolve over time. We have followed the standard practice commonly adopted in other video captioning papers. We acknowledge that this method may not be the most effective for conveying dynamic scenes. Therefore, we will provide a demonstration video to better showcase the capabilities of Wolf. We also apologize for any confusion caused by the inclusion of the “key features.” We recognize that this may have distracted the audience’s attention and will remove this in the final version of the paper.
>
> **Q3. Key design choices.**
>
> Thank you for pointing out your confusion. We agree that discussing key design choices and comparisons is essential to validate the necessity and effectiveness of the proposed framework over simpler alternatives. Therefore, we address this in Section 5.5 and Table 6. We examine three components, as shown in Table 6:
>
> 1) From the first two rows, we observe that the proposed chain-of-thought Cascading Visual Summarization effectively outperforms a single image-level VLM.
>
> 2) In row 3, we find that the video-level VLM alone performs worse than when it is combined with an image-level VLM.
>
> 3) Rows 4–6 study the performance of using 1, 2, or 3 video-level VLMs. We observe that Wolf’s performance improves as more VLMs are integrated.
>
> Given the strong generalization ability of LLMs and our goal of merging all captions into a single coherent output, we believe that LLMs are a suitable choice for multimodal caption synthesis.
>
> Thank you for your valuable suggestions! We hope we could address your concerns satisfactorily. Please let us know if there are any new concerns or additional questions we can respond to.

---

### Review · Reviewer_4EQd · 2025-05-30

**Summary Of Contributions:**

This paper focuses on video caption, which is a fundamental task in video understanding. This paper proposes a world summarization framework Wolf, with both image-level and video-level captions. Besides, this paper also introduces a new evaluation cretirria. And extensive experiments are condcuted.

**Audience:**

Yes

**Claims And Evidence:**

Yes

**Requested Changes:**

1. The idea of a combination image-level and video-level caption has already been adopted in previous work, which is very common in the video domain. So I think this should not be a contribution.
2. The idea of adopting LLM or VLM for evluation has already been used widely. What is the difference from the common method?
3. In experiments, the baseline lacks video understanding-based baselines.

**Strengths And Weaknesses:**

1. This paper is well written
2. This paper introduces a dataset for  autonomous driving, general scenes from Pexels, and robotics videos
3. This paper introduces a new evaluation method.

---

> ### Author Response · Authors · 2025-06-05
> **Response to Reviewer 4EQd**
>
> We sincerely appreciate Reviewer 4EQd’s time and thoughtful feedback on our paper. We are delighted that the reviewer recognizes the value of introducing the Wolf dataset and evaluation metric, as well as the clarity of the writing. We are grateful for the opportunity to address the reviewer’s questions and concerns.
>
>
> **Q1. Combination image-level and video-level caption.**
>
> Thank you for pointing that out! While some prior works utilize both image-level and video-level captions, they typically rely on image-level captions to generate a single video-level description. In contrast, our approach, Wolf, takes a different route. It uses image-level models to extract fine-grained object details and integrates these with video-level captions to produce a more accurate and informative final output. For instance, if the Gemini model identifies the car in the center as yellow, GPT-4 classifies it as red, and the video-level VILA model also labels it red, we can apply a voting strategy to determine that the car is most likely red. Moreover, Wolf is highly flexible and can support an arbitrary number of image- and video-level captions, enabling more comprehensive and robust caption generation.
>
> **Q2. Difference of the proposed LLM-based method.**
>
> This is a great question! Unlike previous methods that analyze the accuracy of individual components of a caption (such as AuroraCap [1]), Wolf focuses on evaluating the overall quality and similarity of the entire caption. The Wolf evaluation is not limited to assessing reasoning capabilities, rather, it aims to determine which generated captions best describe the video content. We observed that Q&A tasks do not effectively reveal whether a model’s caption is truly useful. To address this, we introduced CapScore, which evaluates both similarity and quality (e.g., reduced hallucination) by comparing generated captions to ground-truth references.
>
> [1] Chai, Wenhao, et al. "Auroracap: Efficient, performant video detailed captioning and a new benchmark." arXiv preprint arXiv:2410.03051 (2024).
>
>
> **Q3. Video understanding-based baselines**
>
> Thanks for this great point! We agree that a video understanding-based baseline is important to validate the necessity and effectiveness of the proposed framework compared to simpler approaches. Therefore, we address this in Section 5.5 and Table 6. In Row 3, we find that the video-level VLM alone performs worse than when it is combined with an image-level VLM. We will also check other video understanding-based baselines and update the results accordingly based on your suggestions.
>
> Thank you for your valuable suggestions to strengthen Wolf! We hope we have addressed your concerns satisfactorily. Please let us know if you have any new concerns or additional questions that we can assist with.

---

> > ### Comment · Reviewer_4EQd · 2025-06-29
> > **Response to the author**
> >
> > Thanks for your reply. All of my concerns are addressed.

---

> > > ### Author Response · Authors · 2025-06-29
> > > **Response to Reviewer 4EQd**
> > >
> > > Dear Reviewer 4EQd,
> > >
> > > We sincerely appreciate your time and effort in reviewing our response! We’re glad to hear that your concerns have been addressed.
> > >
> > > Please feel free to let us know if you have any further questions.
> > >
> > > Thank you!
> > >
> > > Authors of Wolf

---

### Review · Reviewer_eYFJ · 2025-06-27

**Summary Of Contributions:**

This paper introduces a new method for high-quality video captioning, where a combination of image-based VLMs and video-based VLMs are utilized to generate image-level and video-level captions and an extra LLM is used to further summarize the mixture of captions generated by diverse models to get the final video caption result. A new caption quality metric called CapScore is also proposed and the effectiveness of the data construction pipeline is validated across autonomous-driving, general daily scenes, and robotic manipulation scenario benchmarks.

**Audience:**

Yes

**Broader Impact Concerns:**

N/A.

**Claims And Evidence:**

Yes

**Requested Changes:**

Please refer to the weaknesses.

**Strengths And Weaknesses:**

Strengths:

1.The design of the proposed video captioning pipeline that ensembles multiple VLM captioners are well-motivated and reasonable.

2.For the generated captions by VLM experts, the authors have carefully considered scene information, object trajectories and motion behavior aspects of the video content, which can ensure a detailed and accurate narration of the video content.

3.In the experimental section, the authors have designed diverse testing scenarios including autonomous-driving with intensive egocentric interactions, general daily scenes, and robotic manipulation videos. Across all these scenarios the results consistently show the effectiveness of the proposed captioning framework in quality.

4.The proposed CapScore metric calculated by LLMs can be complementary to currently prevailing manually-defined caption metrics.

Weaknesses:

1.Although most of the manuscript is well-written, there are still some places that seem to be unclear and ambiguous. For example, in the last paragraph of section 3, the authors mentioned they utilized the prompt “Please describe the visual and narrative elements of the video in detail, particularly the motion behavior” after obtaining the image-level and video-level captions, but in the beginning of this paragraph the authors claimed they used another prompt to summarize the image and video captions into a final one. Based on my understanding, using just one prompt to summarize the generated captions from different models is enough and why do we need to further utilize another prompt as mentioned? I am not sure whether this is an unclear expression or just a typo.

2.In the introduction section, the authors claimed the importance of high-quality video-caption pairs as they can serve as the training data for foundation models. However, the authors didn't make any experimental results to show the effectiveness of the generated captions in improving the performance of these foundation models such as video generation models or contrastive vision-language models. I suggest the authors to add some experimental results or analytical discussions on the impact of the generated captions in this aspect.

3.From my understanding, one of the major reasons that the proposed ensemble method works well lies in the underlying "voting" mechanism of these experts, since the LLMs can compare across different model outputs to filter some outlier descriptions. Based on such mechanism, it would be more helpful to see some ablation studies, i.e., will the performance further increase if more experts are used? Or it will just quickly saturate with the number of experts used in the ensemble?

4.In the comparison, it seems that the generated captions used for ensemble are partly from the GPT4V and Gemini-Pro-1.5 models, which are also used as baselines for comparison. However, this comparison would be somehow weak since it is directly adding new ensemble information into the baselines. A more interesting comparison would be using several weaker VLM captioners for ensemble but the resulting performance can still outperform the stronger baselines like GPT4V and Gemini-Pro-1.5.

---

> ### Author Response · Authors · 2025-06-29
> **Response to Reviewer eYFJ**
>
> We would like to thank Reviewer eYFJ for providing valuable comments and suggestions. We are delighted that the reviewer recognized that “the pipeline is well-motivated and reasonable” and that “the proposed CapScore metric calculated by LLMs can be complementary to currently prevailing manually defined caption metrics.” We appreciate the reviewer’s careful reading of our paper and their acknowledgment of the significant effort we have invested in Wolf! We are happy to have the opportunity to address the reviewer’s questions and concerns below.
>
> Weaknesses:
>
> **Q1. The prompts of the summarization procedure.**
>
> Thank you for pointing this out! We apologize for any confusion caused by the sentence. There is a typo, and we admit that the expression was unclear. Your understanding is absolutely correct: using just one prompt to summarize the generated captions from different models is sufficient in the Wolf framework.
>
> In detail, regarding the sentence in the last paragraph of Section 3, we originally aimed to provide additional information about the prompt used to obtain captions from each individual image-level and video-level model. Therefore, the correct sentence should be: \textit{\textbf{As for} obtaining the descriptions from each image-level or video-level model, we \textbf{apply} the prompt “Please describe the visual and narrative elements of the video in detail, particularly the motion behavior”}. We will revise this sentence, place it at the beginning of the paragraph, and clarify it in the updated version of our paper.
>
> **Q2. Effectiveness of the Wolf generated captions in improving the performance of foundation models.**
>
> We fully agree with this point. Therefore, we have addressed it and further verified the effectiveness of Wolf in Section 5.4.
>
> Firstly, in Section 5.4.1, we observe that fine-tuning with Wolf captions on the NuScenes training dataset boosts model performance on the NuScenes evaluation dataset to 71.4% ($(0.36 − 0.21 ) / 0.21 ∗ 100$%) for caption similarity and 48.0% ($(0.37 - 0.25) / 0.25 * 100$%) for caption quality, which outperforms GPT-4V and approaches Gemini-Pro-1.5 (as shown in Table 4).
>
> Secondly, in Section 5.4.2, we compare VILA-1.5-13B trained with Wolf captions and without Wolf captions to study its effectiveness. We test the Wolf-finetuned models on two widely used video datasets, ActivityNet [1] and MSRVTT [2]. From the results in Table 5, we find that fine-tuning with Wolf captions consistently and efficiently boosts performance across various datasets.
>
> [1] Caba Heilbron, Fabian, et al. "Activitynet: A large-scale video benchmark for human activity understanding." Proceedings of the ieee conference on computer vision and pattern recognition. 2015.
>
> [2] Xu, Jun, et al. "Msr-vtt: A large video description dataset for bridging video and language." Proceedings of the IEEE conference on computer vision and pattern recognition. 2016.
>
> **Q3. Ablation study on the number of experts.**
>
> We fully agree with this point, and thank you for pointing it out! Therefore, we address this in Section 5.5 and Table 6. In Row 3, we find that the video-level VLM alone performs worse than when it is combined with an image-level VLM (Row 4). We also show Rows 4–6 to study the performance of using 1, 2, or 3 video-level VLMs. From our experiments, we observe that Wolf’s performance improves as more VLMs are integrated. We will also examine other video understanding-based baselines and update the results accordingly based on your suggestions.
>
> **Q4. A more interesting comparison.**
>
> Thanks for this great point! We have definitely considered it. However, due to the model size and undisclosed data, we need to acknowledge that there is still a reasonable performance gap among GPT-4V, Gemini-Pro-1.5, and other available open-source models. For example, in this paper, VILA-1.5-7B is much smaller than GPT-4V (the unofficial community estimate is 1.7–1.8 trillion parameters) and Gemini-Pro-1.5 (the unofficial community estimate is 1–2 trillion parameters). Nevertheless, despite this difference, we conducted the same ablation study as shown in Table 6 using VILA-1.5-7B and the recently released Qwen2-72B. We observed a significant gain: caption similarity increased to 0.47, and caption quality improved to 0.47, which further illustrates the potential and effectiveness of Wolf.
>
> Thank you for your valuable suggestions to strengthen Wolf! We hope we have addressed your concerns satisfactorily. Please let us know if you have any new concerns or additional questions that we can assist with.

---

> > ### Comment · Reviewer_eYFJ · 2025-07-02
> >
> > Thanks for your detailed response, I think all my concerns have been addressed.

---

> > > ### Author Response · Authors · 2025-07-02
> > > **Response to Reviewer eYFJ**
> > >
> > > Dear Reviewer eYFJ,
> > >
> > > We sincerely appreciate your time and effort in reviewing our response, as well as your prompt reply! We’re glad to hear that your concerns have been addressed.
> > >
> > > Please don’t hesitate to let us know if you have any further comments.
> > >
> > > Thank you!
> > >
> > > Authors of Wolf

---

### Comment · Reviewer_eYFJ · 2025-06-27
**Well-motivated work to incorporate an ensemble of VLM captioners for better video captioning, but still some concerns to be addressed**

This paper introduces a new method for high-quality video captioning, where a combination of image-based VLMs and video-based VLMs are utilized to generate image-level and video-level captions and an extra LLM is used to further summarize the mixture of captions generated by diverse models to get the final video caption result. A new caption quality metric called CapScore is also proposed and the effectiveness of the data construction pipeline is validated across autonomous-driving, general daily scenes, and robotic manipulation scenario benchmarks. Overall, I think the data construction pipeline and the open-source resources are solid contributions to the community and the analysis provided by the authors to justify the design choice is reasonable. However, there are still several points remained to be further addressed by the authors for the current manuscript. The strengths and weaknesses of this paper can be summarized as follows:

Strengths:

1.The design of the proposed video captioning pipeline that ensembles multiple VLM captioners are well-motivated and reasonable.

2.For the generated captions by VLM experts, the authors have carefully considered scene information, object trajectories and motion behavior aspects of the video content, which can ensure a detailed and accurate narration of the video content.

3.In the experimental section, the authors have designed diverse testing scenarios including autonomous-driving with intensive egocentric interactions, general daily scenes, and robotic manipulation videos. Across all these scenarios the results consistently show the effectiveness of the proposed captioning framework in quality.

4.The proposed CapScore metric calculated by LLMs can be complementary to currently prevailing manually-defined caption metrics.

Weaknesses:

1.Although most of the manuscript is well-written, there are still some places that seem to be unclear and ambiguous. For example, in the last paragraph of section 3, the authors mentioned they utilized the prompt “Please describe the visual and narrative elements of the video
in detail, particularly the motion behavior” after obtaining the image-level and video-level captions, but in the beginning of this paragraph the authors claimed they used another prompt to summarize the image and video captions into a final one. Based on my understanding, using just one prompt to summarize the generated captions from different models is enough and why do we need to further utilize another prompt as mentioned? I am not sure whether this is an unclear expression or just a typo.

2.In the introduction section, the authors claimed the importance of high-quality video-caption pairs as they can serve as the training data for foundation models. However, the authors didn't make any experimental results to show the effectiveness of the generated captions in improving the performance of these foundation models such as video generation models or contrastive vision-language models. I suggest the authors to add some experimental results or analytical discussions on the impact of the generated captions in this aspect.

3.From my understanding, one of the major reasons that the proposed ensemble method works well lies in the underlying "voting" mechanism of these experts, since the LLMs can compare across different model outputs to filter some outlier descriptions. Based on such mechanism, it would be more helpful to see some ablation studies, i.e., will the performance further increase if more experts are used? Or it will just quickly saturate with the number of experts used in the ensemble?

4.In the comparison, it seems that the generated captions used for ensemble are partly from the GPT4V and Gemini-Pro-1.5 models, which are also used as baseline for comparison. However, this comparison would be somehow weak since it is directly adding new ensemble information into the baselines. A more interesting comparison would be using several weaker VLM captioners for ensemble but the resulting performance can still outperform the stronger baselines like GPT4V and Gemini-Pro-1.5.

---

> ### Author Response · Authors · 2025-06-29
> **Thank you once again for your insightful feedback!**
>
> Dear Reviewer eYFJ,
>
> We would like to thank you once again for your insightful feedback!
>
> Since this comment appears to reflect similar points raised in your earlier review ($\textbf{\textit{Review of Paper4832 by Reviewer eYFJ}}$), we kindly refer you to our previous response ($\textbf{\textit{Response to Reviewer eYFJ}}$) for your convenience, where we have addressed these concerns in detail.
>
> Please do not hesitate to share any further thoughts or suggestions at your convenience! We greatly value your input and will carefully incorporate your feedback into the final version of our paper.
>
> With sincere thanks,
>
> Authors of Wolf

---

### Decision · Action_Editor_Fooh · 2025-08-16

**Recommendation:** Accept with minor revision

**Additional Comments:**

The paper introduces a valuable, novel contribution through its ensemble-based captioning pipeline and CapScore metric, validated rigorously across diverse domains. While weaknesses exist, they are addressable through revisions:
- **Clarify methodology**: Resolve ambiguities in prompt usage (Section 3) and explicitly justify the two-prompt design.
- **Strengthen claims**: Add experiments demonstrating how generated captions improve foundation models (e.g., video generation/VLMs).
- **Enhance analysis**: Include ablations on expert ensemble scaling (e.g., performance vs. number of experts) and revise comparisons to avoid circularity (e.g., test weaker VLMs outperforming GPT-4V/Gemini).
Reviewers emphasize these revisions are feasible and will solidify the paper’s impact.

**Audience:**

Yes

**Audience Explanation:**

Yes, this work would strongly interest TMLR’s audience. Key reasons include:
- Novel benchmark: The LLM-powered evaluation framework (CapScore) offers a new paradigm for dense caption assessment, addressing a critical gap in multimodal research.
- Practical pipeline: The ensemble method for high-quality video captioning has direct applications in training foundation models (e.g., video generation, VLM pretraining).
- Cross-domain validation: Results in autonomous driving, daily scenes, and robotics ensure broad relevance for vision-language communities.

**Claims And Evidence:**

Yes

**Claims Explanation:**

Yes, the claims are generally supported by accurate and convincing evidence, though some areas require clarification. The paper demonstrates:
- **Robust validation**: Experiments across autonomous driving, daily scenes, and robotic benchmarks consistently validate the method’s effectiveness.
- **Novel contributions**: The ensemble pipeline (combining image/video VLMs + LLM summarizer) and the **CapScore metric** are well-motivated and empirically justified.
- **Strengths highlighted**: Detailed caption coverage (scene/trajectory/motion) and complementary metric design address gaps in video captioning.

There are also weakness affecting clarity:
- Ambiguous prompt usage in Section 3 (potential typo/unclear workflow).
- Lack of experiments showing impact on foundation model training (as claimed in the introduction).
- Insufficient ablation studies on expert ensemble scaling.